# A Retrospective Study Comparing Outcomes of Paravertebral Clonidine Infusion for Pediatric Patients Undergoing Nuss Procedure

**DOI:** 10.3390/children10020193

**Published:** 2023-01-20

**Authors:** Mihaela Visoiu, Senthilkumar Sadhasivam

**Affiliations:** Department of Anaesthesiology and Perioperative Medicine, University of Pittsburgh Medical Center, UPMC Children’s Hospital of Pittsburgh, 4401 Penn Avenue, Pittsburgh, PA 15224, USA

**Keywords:** Nuss procedure, pectus excavatum, paravertebral block catheters, clonidine, postoperative pain control

## Abstract

Introduction: The continuous paravertebral blockade as part of the multimodal pain protocol is an effective regional technique to control pain after the Nuss procedure. We investigated the effectiveness of clonidine as an adjunct to paravertebral ropivacaine infusion. Methods: We conducted a retrospective study of 63 patients who underwent Nuss procedures and received bilateral paravertebral catheters. Data on demographics, surgical, anesthesia, and block characteristics, numeric rating pain scores (NRS), opioids consumption, hospital length of stay, complications, and side effects from medication administration were collected in children who received paravertebral ropivacaine 0.2% infusion without (N = 45) and with clonidine (1 mcg/mL) (N = 18). Results: The two groups had similar demographics, although the clonidine group had higher Haller indices (6.5 (4.8, 9.4) vs. 4.8 (4.1, 6.6), *p* = 0.013). The clonidine group required less morphine equivalent/kg on postoperative day 2 (median, interquartile range 0.24 (0.22, 0.31) vs. 0.47 (0.29, 0.61) *p* = 0.002). There was no difference in median NRS pain scores. Both groups had similar catheter infusion durations, hospital length of stay, and complication rates. Conclusion: A postoperative pain management plan that includes paravertebral analgesia, including clonidine as an adjunct, may be considered to minimize opioid administration for patients undergoing primary Nuss repair.

## 1. Introduction

Pectus excavatum is the most common congenital chest wall deformity. It has an estimated incidence of one to eight in 1000 births [1]. Pectus excavatum can cause severe psychological and physiological impairments on patients’ quality of life due to body image issues and cardiovascular and pulmonary limitations on physical activity [2]. The most common surgical technique for correcting pectus excavatum is the Nuss technique [3]. Despite its minimally invasive approach, the Nuss procedure has been associated with significant postoperative pain that can be very challenging to control due to the stretch and pressure on the chest wall after the repair [4].

There remains to be more consensus on managing postoperative pain in this surgical population. Multiple modalities, including thoracic epidurals, paravertebral nerve blocks, wound catheters, multimodal management, intrathecal morphine, erector spinae catheter blocks, and cryotherapy, have been reported at different institutions [4,5,6,7,8,9,10]. The Society of Pediatric Anesthesia Improvement Network (SPAIN) published a benchmark study describing the various techniques used by fourteen institutions for perioperative and postoperative pain management in patients undergoing Nuss repair [11]. However, only two institutions performed paravertebral nerve blocks for their patients [11]. At UPMC Children’s Hospital of Pittsburgh, the bilateral paravertebral nerve block catheters were the primary regional anesthesia technique for the Nuss procedures. We used a multimodal analgesia protocol involving paravertebral nerve block continuous infusion with ropivacaine 0.2% and patient control analgesia (PCA) with hydromorphone (continuous infusion and demand). In 2016, we piloted a protocol that involved adding clonidine as an adjunct to ropivacaine infusion and removing continuous hydromorphone infusion. These two pain management techniques are evolutions of each other. We hypothesized that adding clonidine as an adjunct to ropivacaine for continuous paravertebral blocks allows the removal of PCA continuous infusion without an increase in pain scores. In addition, this change will decrease opioid consumption, hospital stay, complications, and side effects from medication administration.

## 2. Materials and Methods

This retrospective study was conducted at UPMC Children’s Hospital of Pittsburgh from September 2014 to July 2018. It was approved by the University of Pittsburgh Institutional Review Board that waived requirements for informed consent. Patients who underwent a Nuss procedure for pectus repair and received bilateral paravertebral catheter and a multimodal approach for postoperative pain management were included in the study. The following patients were excluded: (1) patients who underwent the Ravitch procedure, (2) repeat revision of a prior Nuss repair, (3) patients older than age 18 years, and (4) patients with a chronic pain history and/or requiring opioid pain medications for more than a month before surgery.

Data were collected from the patients’ electronic medical records. The primary objectives of the study were to compare: (1) daily numerical rating scale (NRS) pain scores, (2) total daily opioid consumption in morphine equivalents, and (3) length of catheter infusion duration and hospital stay in days. The secondary objective was to compare complications associated with nerve block placement and/or side effects of medication.

### 2.1. Protocol Description

All surgeries were performed by experienced surgeons using a standardized surgical procedure. Before surgery, the bilateral ultrasound-guided thoracic paravertebral nerve catheters were placed at the T4-T7 level under sedation or general anesthesia, as described by Borestky and et al. [12]. A bolus of 10–15 mL of 0.5% ropivacaine was administered through each catheter after placement (dose not to exceed 3 mg/kg). Anesthesia induction and maintenance were similar, with midazolam, propofol, sevoflurane, fentanyl, hydromorphone, morphine, ketamine, acetaminophen, dexamethasone, dexmedetomidine, and muscle relaxant for tracheal intubation. Ketorolac was administered if the surgeon agreed.

### 2.2. Old Protocol (Paravertebral Catheter Infusion with Ropivacaine 0.2%)

On POD 0, as soon as the patient arrived in the patient anesthesia care unit (PACU), local anesthetic infusions through the bilateral paravertebral nerve catheters were initiated with 0.2% ropivacaine at a rate of 8 mL/h on each side, with 2 mL clinician boluses administered by the nurse every 180 min on each side as needed for each side. Each patient was given hydromorphone via PCA, which included a demand dose of 5 mcg/kg every 10 min, a clinician bolus of 7 mcg/kg every 30 min, and a continuous infusion at 1 to 2 mcg/kg/h. The ketamine infusion was started at 0.1 mg/kg/h. Other medications administered were intravenous (IV) acetaminophen 15 mg/kg every 6 h (maximum 4000 mg/day), oral (PO) gabapentin 300 mg every night (unless the patient was sedated), IV diazepam 0.05 mg/kg every 6 h as needed for muscle spasms, and IV ondansetron 4 mg every 6 h. Ketorolac was administered if the surgeon agreed.

On POD 1, the paravertebral nerve catheter infusions were adjusted as necessary to improve pain control. The hydromorphone PCA was continued without the continuous infusion, while adjustments were made to the demand dose and clinician bolus. The ketamine infusion, IV acetaminophen, PO gabapentin, and IV diazepam were continued. Cyclobenzaprine 5 to 10 mg twice daily was added to ketorolac 0.5 mg/kg (maximum dose 15 mg) every 6 h (after surgical approval).

On POD 2, the paravertebral nerve catheters were continued. The hydromorphone PCA was discontinued. Oxycodone 0.1 mg/kg, maximum 5 mg, every 4 h as needed, and IV hydromorphone, 5 mcg/kg, available for breakthrough pain every 2 h were started. The acetaminophen was switched from the IV route to the PO route. The gabapentin, diazepam, cyclobenzaprine, and ketorolac were continued. The scheduled ondansetron was changed to as needed every 6 h.

On POD 3, the paravertebral nerve catheter infusions were continued. If the hydromorphone PCA was not discontinued the day prior, it was discontinued on POD 3. The oxycodone was adjusted to 0.15 to 0.2 mg/kg, a maximum of 10 mg/dose, every 4 h as needed for pain control. The ketamine infusion was discontinued, while the acetaminophen, gabapentin, diazepam, and cyclobenzaprine were continued. The ketorolac was switched to ibuprofen 4–10 mg/kg every 6 h with a maximum dose of 3200 mg/daily. The catheters were suspended and removed if the patient’s pain was controlled well.

On POD 4, the paravertebral nerve catheter infusions were suspended at 6 a.m. and removed at 10 a.m. if the patient’s pain was controlled with the other multimodal analgesics. The acetaminophen, cyclobenzaprine, ibuprofen, and oxycodone were continued until the patient’s discharge home.

### 2.3. The New Protocol (Paravertebral Catheter Ropivacaine and Clonidine Infusion)

At the end of the surgery, 0.5 mg/kg of ketorolac (max dose 15 mg) was administered unless the surgeon contraindicated.

On POD 0, infusions through the bilateral paravertebral nerve catheters were initiated with 0.2% ropivacaine with 1 mcg/mL of clonidine at a rate of 8 mL/h on each side, with 2 mL clinician boluses administered by the nurse every 180 min on each side as needed. Each patient was also given a hydromorphone PCA, with the same demand and clinician bolus amount but without continuous infusion. Ketorolac was continued at 0.5 mg/kg (maximum dose 15 mg) every 6 h. The rest of the medications for pain control (ketamine infusion, acetaminophen, diazepam) and to prevent nausea were similar. Gabapentin was not started.

On POD 1, the paravertebral nerve catheter infusions were adjusted as necessary to improve pain control. The hydromorphone PCA was discontinued at the end of the day. Oxycodone 0.1 mg/kg every 6 h and IV hydromorphone 5 mcg/kg every 2 h for breakthrough pain were started. Gabapentin was administered only in the case it was indicated for pain control. The rest of the medications for pain control and to prevent nausea were similar.

On POD 2, the paravertebral nerve catheters were continued. The ketamine infusion was discontinued. In addition to the previous opioids, 5 mg of as-needed oxycodone every 4 h was added. The rest of the medications for pain control and to prevent nausea were similar. The catheters were suspended and removed if the patient’s pain was controlled well.

On POD 3, the paravertebral nerve catheter infusions were suspended at 6 a.m. and removed 4 h later if the patient’s pain continued to be well controlled. The oxycodone could also be adjusted to 0.15 mg/kg every 4 h for pain, a maximum of 10 mg/dose. The rest of the medications for pain control and to prevent nausea were similar.

On POD 4, if the paravertebral nerve catheters were still in situ, they were suspended and removed.

A few differences between these two protocols facilitated a quicker transition from the PCA at the end of POD 1 rather than on POD 2 and the removal of paravertebral catheters POD 3 rather than POD 4. These included adding clonidine to the paravertebral nerve catheters, removing the continuous infusion on the hydromorphone PCA, and starting ketorolac in POD 0. The addition of scheduled oxycodone with as-needed supplemental oxycodone facilitated the discontinuation of the ketamine infusion on POD 2 rather than POD 3.

### 2.4. Analyses

Outcomes were analyzed by postoperative day with POD 0 defined as the end of surgery to midnight of the same evening. POD 1 started at midnight after surgery and extended for 24 h. The data were collected in this manner until discharge. Total intravenous and enteral opioid consumption was collected for each POD and converted to morphine equivalents using the opioid equivalency table used by the *Practical Pain Management Journal* (https://opioidcalculator.practicalpainmanagement.com (accessed on 15 January 2023)).

Descriptive statistics were summarized as frequencies (percentages, %) for categorical data or as the median and interquartile range for non-normally distributed continuous data. Examination of normal distribution assumption for continuous data was determined by q-q plots and histograms. Wilcoxon–Mann–Whitney test was performed to determine differences between groups for non-normally distributed continuous data, respectively. Pearson’s chi-square or Fisher’s exact test, as appropriate, was used to compare the frequency distribution of categorical variables between the groups. For NRS pain scores and morphine equivalents of opioid consumption data, linear mixed-effects models were used to test the main effects of group (old protocol versus new protocol), time (POD 0 to POD 6 for pain scores; POD 1 to POD 6 for morphine equivalents of opioid consumption) and time by group interactions and to account for within-subject correlation. One between-subjects factor (group) and one within-subjects factor (time; POD) and their interaction were defined as fixed effects and the subject as a random effect. For NRS pain scores, the analyses were adjusted by NRS pain scores during PACU. For linear mixed-effects models, logarithm base 10 transformations were used for morphine equivalents of opioid consumption data. After inspecting the correlation within subjects, an autoregressive structure was assumed. Due to discrete medication doses, low number (N) comparisons, and low-frequency administration of certain drugs, interquartile ranges (IQRs) may show all zeros. This is not an error and shows that at least three-quarters of the comparable sample received no dose and suggested uncommon medication used in this study. All analyses were two-sided, and the significance level was set to 0.05. The data were analyzed using SAS (version 9.3; SAS Institute, Cary, NC, USA).

## 3. Results

### 3.1. Demographics

Sixty-three patients were included in the study (Table 1). Forty-five patients were exposed to the old protocol, while eighteen patients were exposed to the new protocol. The two groups were similar in their characteristics, including age, sex, weight, number of bars placed, and ASA classification (Table 1).

The main differences between the two groups included a higher Haller index in the new protocol group (6.5 (4.8, 9.4)) compared with the old protocol group (4.8 (4.1, 6.6), *p* = 0.013). The new protocol group had a higher percentage of patients undergoing general anesthesia for the placement of nerve blocks (72% general anesthesia vs. 28% awake/sedation). In comparison, the old protocol group had a higher percentage of awake patients with sedation for nerve block placements (33% general anesthesia vs. 67% awake/sedation, *p* = 0.010). The new protocol group had longer anesthesia times (204.0 (177.0, 238.0) min vs. 178.0 (163.0, 207.0) min, *p* = 0.057) and surgical times (96.5 (83.0, 134.0) min vs. 79.0 (68.0, 97.0) min, *p* = 0.02) compared with the old protocol group.

#### 3.1.1. Pain Scores

There was no difference in median NRS pain scores between the two groups on all postoperative days, although the pain scores were lower in the new protocol group (Figure 1).

#### 3.1.2. Analgesic Administration

There was no significant difference in administered medications between the two groups, except ketorolac being more commonly administered intraoperatively in the new protocol group compared with the old protocol group (six patients (33%) in the new protocol group vs. five patients (11%) in the old protocol group, *p* = 0.068) (Table 2). The amount of ropivacaine mg/kg administered before the catheter placement was 2.60 (2.10, 3.00) (old protocol) vs. 2.35 (1.60, 3.60) (new protocol). The medications administered intraoperatively are presented in Table 2. There was no difference in total morphine equivalents (median/interquartile range, mg) administered intraoperatively in PACU and on POD 0, 1, 2, 3, 4, 5 (Figure 2). The medications administered in PACU and postoperative days 1–6 are presented in Table 3 and Table 4, respectively. With the new protocol (clonidine group), the patient required less morphine equivalent/kg on postoperative day 2 (median, interquartile range 0.24 (0.22, 0.31) vs. 0.47 (0.29, 0.61) *p* = 0.002) (Table 5).

#### 3.1.3. Postoperative Outcomes

There were no differences in catheter infusion duration and hospital length of stay (Table 1).

#### 3.1.4. Complications

The Table 6 shows the number and percentage of complications in the study. The most common complications included muscle spasms, nausea, and pruritis. However, there were no differences in the number of complications, types, and medication side effects between the new and old protocol groups.

## 4. Discussion

Clonidine is known to act on α_2_ adrenoreceptors, but some studies suggest alternative mechanisms of action and target sites. It is described as an adjuvant to local anesthetics [13] and it is indicated for use with local anesthetic for single-injection peripheral nerve blocks [4,5,6,7,8,9,10,11,12,13,14,15,16,17,18,19].

Pain after the Nuss procedure is challenging to treat; ultrasound-guided continuous paravertebral blocks can be considered for pain control. In this cohort study, 18 patients with bilateral continuous paravertebral blocks received a combination of ropivacaine 0.2% and clonidine 1 mcg/mL. They were compared with 45 patients who received only ropivacaine 0.2% infusion. The clonidine group required less morphine equivalent/kg on a postoperative day 2 (median, interquartile range 0.24 (0.22, 0.31) vs. 0.47 (0.29, 0.61) *p* = 0.002). This almost 50% reduction in opioid requirement on POD 2 is clinically significant because the clonidine group had higher Haller indices (6.5 (4.8, 9.4) vs. 4.8 (4.1, 6.6), *p* = 0.013). Higher Haller indices indicate severe pectus repair and more postoperative pain to correct severe surgical pain.

Despite the minute changes we made between our old and new protocols within our institution (including the addition of multimodal medications such as ketorolac, clonidine in the nerve block and discontinuation of continuous infusion of intravenous opioid), there was very little difference seen in pain scores and opioid consumption between the two groups. The decrease in opioid consumption on POD 2 may reflect overall changes in the new protocol, which included adding clonidine to the paravertebral infusion. Despite removing continuous infusion of opioids, there was no increase in the pain scores on the day of surgery and PCA hydromorphone was discontinued on POD 1. Perhaps our results were not more pronounced because of the significantly higher Haller indexes in the clonidine group, which added a higher pain burden that was better treated with clonidine.

Other studies show improved pain control and decreased opioid consumption with clonidine in a paravertebral nerve block [20,21]. Mayur and al. found that clonidine used as an adjunct in thoracic paravertebral single injection blocks provided profound analgesia for up to 48 h [21]. Bhatnagar et al. found that patients who received 0.125% bupivacaine 2 mg/kg with clonidine 2 mcg/kg via paravertebral single injection, followed by paravertebral infusion of 0.125% bupivacaine at 0.5 mg/kg/h, with clonidine at 2 mcg/kg/h had reasonable pain control, but a higher incidence of hypotension. We considered adding clonidine to the single paravertebral injections (1–2 mcg/kg), but we noticed that patients tend to be hypotensive during the surgery and in PACU. We added less clonidine to our paravertebral infusion than Bhatnagar did. We did not collect data to compare the incidence of hypotension and bradycardia between the two groups. Still, we did not have any instances where we needed to discontinue paravertebral infusion because of hypotension or bradycardia.

According to the Society for Pediatric Anesthesia Improvement Network (SPAIN) study, only two out of fourteen institutions performed paravertebral nerve blocks for Nuss patients. In contrast, the majority performed thoracic epidurals [11]. None of these institutions use clonidine as an adjunct for paravertebral infusion. While epidurals resulted in less opioid consumption than other modalities, pain scores were similar. Benefits of paravertebral nerve blocks include fewer hemodynamic side effects, no foley requirement, lack of motor weakness, and shorter length of stay [11]. When compared with the SPAIN study, our patients, despite a higher Haller index, showed similar pain scores, postoperative opioid consumption, and length of stay. The opioid consumption was significantly lower on POD 2, and our study might suggest that paravertebral clonidine infusion may add some benefit to postoperative pain control with minimal additional risk.

The lack of difference in nerve catheter infusion duration and hospital length of stay duration can be explained by the inherent post-surgical course, specifically the presence of the chest tube. Another explanation can be related to our initial informal goal of suspending the paravertebral catheters on POD 3 in the old protocol that was subsequently formalized in the new protocol, therefore not much difference in nerve catheter duration was seen. After that, because the nerve catheter must be discontinued before discharge with the assurance of continued pain control on an oral pain regimen, it is not surprising that the hospital stay is about 24 h longer than the nerve catheter infusion duration.

Recently, intercostal nerve cryoablation (INC) has been performed to improve pain control after the Nuss procedure [4]. This procedure decreased hospital opioid consumption, and hospital stays, and many hospitals stopped performing regional anesthesia techniques. Our daily opioid consumption in the clonidine group is less than the regimen that included INC (0.32 mg/kg (0.15–0.54). The patients who underwent INC could develop prolonged numbness and neuropathic pain. It is unclear whether cryoablation provides an increased benefit or decreased risks compared with available regional anesthetic techniques. Paravertebral analgesia should be considered an alternative to INC. The ability of cryoneurolysis to control pain on postoperative day 0–1 has to be investigated more [4], and continuous paravertebral blocks could be considered to improve analgesia on the first two postoperative days when opioid requirements are still high [9,22].

Our study is limited by its retrospective nature. Patients were not randomly assigned to different treatment protocols or a control group; thus, it is difficult to assess the actual degree of differences in pain scores and opioid consumption as well as hospital length of stay. Our limited sample size leads to further limitations in detecting more significant differences between the two protocols. In addition, we did not evaluate essential functional outcomes (pain at rest, with activity, sleeping pattern) in the perioperative period or the incidence of hypotension or bradycardia. Secondly, we did observe changes in practice patterns shortly after completing this study, and a tendency toward using INC increased over the last years. In our institution, continuous paravertebral nerve blocks with ropivacaine 0.2% and clonidine 1 mcg/mL are performed for a group of patients who undergo INC. PCA with hydromorphone was removed from our multimodal pain protocol for pediatric patients undergoing the Nuss procedure.

## 5. Conclusions

The addition of clonidine (1 mcg/mL) to bilateral paravertebral ropivacaine 0.2%, 8 mL/h, as part of a multimodal analgesia approach reduced postoperative opioid requirement after the Nuss procedure. Additional large prospective studies are needed to fully evaluate the efficacy and safety of perineural clonidine infusion in pediatric populations.

## Figures and Tables

**Figure 1 children-10-00193-f001:**
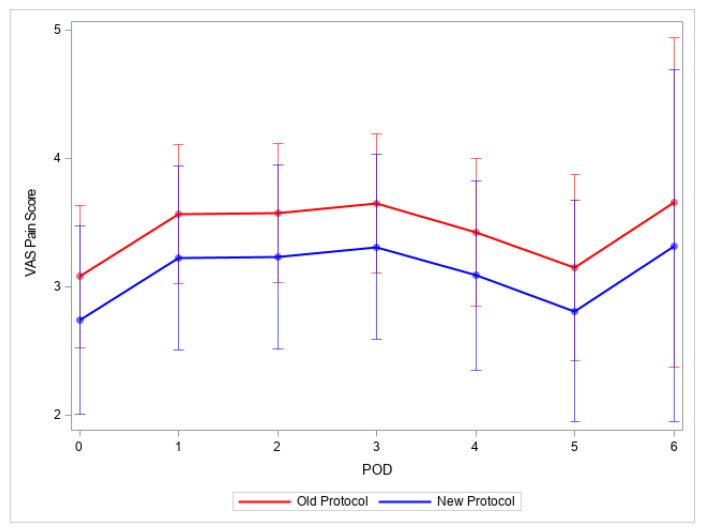
Visual analog scale pain scores (VAS) on postoperative days 0–6. Legend: VAS pain scores were reported as median, interquartile range; POD represents the postoperative day.

**Figure 2 children-10-00193-f002:**
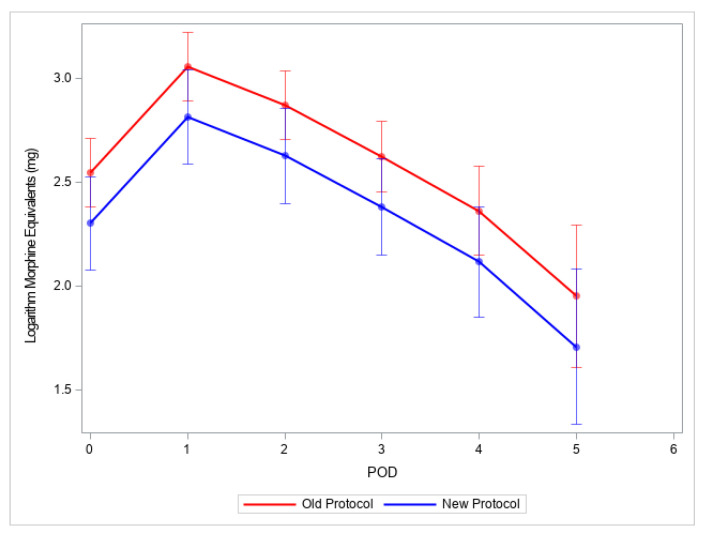
Total morphine equivalents reported in mg on postoperative days 0–5. Legend: POD is the postoperative day; mg is milligrams. Morphine was log-transformed for the analysis.

**Table 1 children-10-00193-t001:** Patient characteristics.

	Median (Q1, Q3) or N (%)		
	Old Protocol/ Ropivacaine	New Protocol/ Ropivacaine Clonidine	*p*-Value
Number of patients	N = 45	N = 18	
Age (years)	15 (14, 16)	15 (14, 16)	0.8740
Gender (male, %)	37 (82)	15 (83)	1.0000
Weight (kg)	56.00 (49.30, 59.40)	52.00 (46.40, 61.90)	0.2130
BMI	18.00 (16.90, 19.80)	18.00 (17.00, 19.00)	0.7340
Haller index	4.80 (4.10, 6.60)	6.45 (4.80, 9.40)	0.0130
Bars			0.7290
1 bar	37 (82)	14 (78)	
2 bars	8 (18)	4 (22)	
ASA class			1.0000
I/II/III	9 (20)/34 (76)/2 (4)	4 (22)/13 (72)/1 (6)	
Surgical service			0.3396
Cardiac surgery	12 (27)	7 (39)	
General surgery	33 (73)	11 (61)	
Nerve block procedure time (min)	22.00 (16.00, 26.00)	20.50 (15.00, 25.00)	0.5446
Nerve block type			
Paravertebral block catheters	45 (100)	18 (100)	
Nerve block level			0.0639
T4	1 (2)	0 (0)	
T5	33 (73)	8 (44)	
T6	9 (20)	9 (50)	
T7	2 (4)	1 (6)	
Nerve block sedation			0.0104
Awake/sedation	30 (67)	5 (28)	
General anesthesia	15 (33)	13 (72)	
Catheter infusion duration (days)	3.60 (2.90, 3.80)	3.35 (2.80, 4.20)	0.8973
Anesthesia time (min)	178.00 (163.00, 207.00)	204.00 (177.00, 238.00)	0.0569
Surgical time (min)	79.00 (68.00, 97.00)	96.50 (83.00, 134.00)	0.0199
Hospital LOS (days)	4.40 (4.00, 5.10)	4.30 (4.10, 5.10)	0.9637

Legend: Q1, Q3 represent interquartile range; N represents the number of patients in each group; % represents percentage; ASA is American Society of Anesthesia physical status; min is minutes; *p*-value, significance level at 0.05.

**Table 2 children-10-00193-t002:** Medications that were administered intraoperatively.

	Old Protocol Ropivacaine		New Protocol Ropivacaine with Clonidine		*p*-Value
	Frequency N(%)	Amount	Frequency N(%)	Amount	
Morphine equivalents					
mg	15.36 (12.68, 20.00)		14.02 (12.50, 23.04)		
mg/kg	0.30 (0.20, 0.40)		0.30 (0.20, 0.40)		
Fentanyl (mcg)	44 (98)	150.00 (100.00, 200.00)	17 (94)	100.00 (100.00, 150.00)	0.3651
Hydromorphone (mg)	13 (29)	0.00 (0.00, 0.30)	8 (44)	0.00 (0.00, 0.60)	0.1151
Morphine (mg)	10 (22)	0.00 (0.00, 0.00)	2 (11)	0.00 (0.00, 0.00)	-
Dexmedetomidine (mcg)	34 (76)	20.00 (8.00, 20.00)	14 (78)	18.00 (8.00, 20.00)	0.9127
Ketorolac (mg)	5 (11)	0.00 (0.00, 0.00)	6 (33)	0.00 (0.00, 15.00)	0.0683
Ketamine (mg)	39 (87)	50.00 (40.00, 60.00)	12 (67)	37.50 (0.00, 50.00)	0.2382
Acetaminophen (mg)	32 (71)	750.00 (0.00, 870.00)	15 (83)	737.50 (650.00, 850.00)	0.9878
Ondansetron (mg)	43 (96)	4.00 (4.00, 4.00)	17 (94)	4.00 (4.00, 4.00)	0.9001
Diphenhydramine (mg)	2 (4)	0.00 (0.00, 0.00)	1 (6)	-	-
Diazepam (mg)	2 (4)	0.00 (0.00, 0.00)	0 (0)	-	-
Midazolam (mg)	37 (82)	4.00 (3.00, 5.00)	15 (83)	4.00 (2.00, 6.00)	0.9814
Dexamethasone(mg)	32 (71)	6.00 (0.00, 8.00)	12 (67)	4.00 (0.00, 8.00)	0.6207
Initial ropivacaine bolus					
mg		148.00 (100.00, 150.00)		113.50 (90.00, 200.00)	
mg/kg		2.60 (2.10, 3.00)		2.35 (1.60, 3.60)	

Legend: Medications were reported in milligrams (mg) or micrograms (mcg), or mg/kg, as median, interquartile range (IQR). N represents the number of patients in each group; % represents percentage; *p*-value, significance level at 0.05. Summary statistics showing all IQRs of 0 due to discrete medication doses, low N comparisons, and low-frequency administration.

**Table 3 children-10-00193-t003:** Medications administered in PACU.

	Old Protocol Ropivacaine		New Protocol Ropivacaine with Clonidine		*p*-Value
	Frequency N(%)	Amount	Frequency N(%)	Amount	
Total morphine equivalents					
mg		2.01 (0.00, 4.34)		1.15 (0.00, 5.00)	
mg/kg		0.04 (0.00, 0.08)		0.03 (0.00, 0.10)	
Fentanyl (mcg)	20 (44)	0.00 (0.00, 27.70)	9 (50)	11.50 (0.00, 50.00)	0.5862
Hydromorphone (mg)	7 (16)	0.00 (0.00, 0.00)	2 (11)	0.00 (0.00, 0.00)	0.6275
Morphine (mg)	7 (16)	0.00 (0.00, 0.00)	3 (17)	0.00 (0.00, 0.00)	1.0000
Ketorolac (mg)	2 (4)	0.00 (0.00, 0.00)	2 (11)	0.00 (0.00, 0.00)	-
Acetaminophen (mg)	8 (18)	0.00 (0.00, 0.00)	2 (11)	0.00 (0.00, 0.00)	-
Diazepam (mg)	2 (4)	0.00 (0.00, 0.00)	0 (0)	-	-

Legend: Medications were reported in milligrams (mg) or micrograms (mcg), or mg/kg, as median, interquartile range (IQR) N represents the number of patients in each group; % represents percentage; PACU is postoperative acute care; *p*-value, significance level at 0.05. Summary statistics showing all IQRs of 0 due to discrete medication doses, low N comparisons, and low-frequency administration.

**Table 4 children-10-00193-t004:** Medications that were administered on postoperative days 1–6.

		Old Protocol Ropivacaine		New Protocol Ropivacaine with Clonidine	*p*-Value
POD 1	N		N		
Ketorolac (mg)	36	30.00 (30.00, 45.00)	16	45.00 (30.00, 60.00)	0.1153
Acetaminophen (mg)	45	3000.00 (2680.00, 3464.00)	18	2755.50 (2250.00, 4000.00)	0.5341
Ketamine (mg)	43	120.00 (79.20, 144.00)	17	96.00 (60.00, 120.00)	0.1170
Cyclobenzaprine (mg)	37	10.00 (10.00, 20.00)	17	10.00 (5.00, 10.00)	0.0684
Diazepam (mg)	7	2.00 (2.00, 3.00)	5	5.00 (5.00, 5.00)	0.0665
Gabapentin (mg)	43	300.00 (300.00, 300.00)	6	300.00 (300.00, 300.00)	0.3093
POD 2					
Ketorolac (mg)	42	60.00 (45.00, 60.00)	18	30.00 (30.00, 60.00)	0.0012
Acetaminophen (mg)	44	2728.50 (2362.50, 3359.00)	17	2795.00 (2250.00, 3640.00)	0.8916
Ketamine (mg)	41	96.00 (60.00, 120.00)	17	39.60 (35.00, 60.00)	0.0003
Cyclobenzaprine (mg)	42	10.00 (10.00, 20.00)	18	10.00 (10.00, 10.00)	0.0411
Diazepam (mg)	7	2.50 (2.00, 2.50)	6	3.75 (2.50, 6.00)	0.2230
Gabapentin (mg)	42	300.00 (300.00, 300.00)	6	300.00 (300.00, 300.00)	0.1326
Ibuprofen (mg)	5	800.00 (600.00, 800.00)	8	800.00 (400.00, 1000.00)	1.0000
POD 3					
Ketorolac (mg)	37	45.00 (30.00, 60.00)	6	37.50 (30.00, 45.00)	0.6917
Acetaminophen (mg)	45	2600.00 (2400.00, 3044.00)	17	2600.00 (2250.00, 4000.00)	0.5420
Ketamine (mg)	29	58.50 (40.60, 72.00)	4	38.75 (32.25, 52.50)	0.2544
Cyclobenzaprine (mg)	42	10.00 (10.00, 20.00)	17	10.00 (10.00, 15.00)	0.3943
Diazepam (mg)	7	3.00 (2.50, 4.00)	8	4.25 (2.25, 11.50)	0.4946
Gabapentin (mg)	36	300.00 (300.00, 300.00)	8	300.00 (300.00, 350.00)	0.0395
Ibuprofen (mg)	30	800.00 (600.00, 1200.00)	15	1200.00 (800.00, 2000.00)	0.0959
POD 4					
Ketorolac (mg)	11	30.00 (30.00, 60.00)	1	-	-
Acetaminophen (mg)	39	2250.00 (1625.00, 3000.00)	15	2000.00 (1300.00, 3000.00)	0.7868
Ketamine (mg)	6	37.75 (25.00, 60.00)	1	25.00 (25.00, 25.00)	0.6319
Cyclobenzaprine (mg)	37	10.00 (10.00, 15.00)	15	10.00 (5.00, 15.00)	0.4740
Diazepam (mg)	1	-	6	3.25 (2.00, 10.00)	-
Gabapentin (mg)	19	300.00 (300.00, 300.00)	4	300.00 (300.00, 350.00)	0.0509
Ibuprofen (mg)	33	1200.00 (800.00, 1600.00)	14	1400.00 (1200.00, 1600.00)	0.3570
POD 5					
Ketorolac (mg)	4	30.00 (22.50, 45.00)	0	-	
Acetaminophen (mg)	22	2537.50 (1300.00, 3000.00)	7	1950.00 (1500.00, 2600.00)	0.5094
Ketamine (mg)	2	65.75 (27.50, 104.00)	0	-	
Cyclobenzaprine (mg)	20	5.00 (5.00, 10.00)	8	10.00 (5.00, 10.00)	0.7010
Diazepam (mg)	1	-	3	5.00 (2.50, 10.00)	0.4370
Gabapentin (mg)	4	300.00 (300.00, 300.00)	2	350.00 (300.00, 400.00)	0.3374
Ibuprofen (mg)	19	1200.00 (800.00, 1200.00)	8	800.00 (700.00, 1600.00)	0.4204
POD 6					
Ketorolac (mg)	1	30.00 (30.00, 30.00)	0	-	
Acetaminophen (mg)	7	1300.00 (1300.00, 1950.00)	3	1500.00 (1300.00, 1950.00)	0.8121
Cyclobenzaprine (mg)	5	5.00 (5.00, 10.00)	3	5.00 (5.00, 10.00)	1.0000
Ibuprofen (mg)	6	1200.00 (800.00, 1200.00)	3	800.00 (800.00, 1200.00)	0.4774

Legend: Medications were reported in mg, as median, interquartile range (IQR). N represents the number of patients in each group; % represents percentage; POD is the postoperative day. *p*-value, significance level at 0.05. Summary statistics showing all IQRs of 0 due to discrete medication doses, low N comparisons, and low-frequency administration.

**Table 5 children-10-00193-t005:** Total morphine equivalents (mg/kg) administered on postoperative days 1–6.

	N	Old Protocol Ropivacaine (mg/kg)	N	New Protocol Ropivacaine Clonidine (mg/kg)	*p*-Value
Intraoperatively	45	0.30 (0.20, 0.40)	18	0.30 (0.20, 0.40)	
PACU	45	0.04 (0.00, 0.08)	18	0.03 (0.00, 0.10)	
POD 1	45	0.26 (0.18, 0.32)	18	0.21 (0.11, 0.31)	0.3046
POD 2	45	0.47 (0.29, 0.61)	18	0.24 (0.22, 0.31)	0.0026
POD 3	45	0.31 (0.24, 0.43)	18	0.32 (0.22, 0.37)	0.5650
POD 4	45	0.23 (0.14, 0.37)	18	0.22 (0.14, 0.37)	0.6656
POD 5	45	0.00 (0.00, 0.22)	18	0.00 (0.00, 0.22)	0.7678
POD 6	45	0.00 (0.00, 0.00)	18	0.00 (0.00, 0.00)	0.9820

Legend: Medications were reported in mg/kg, as median, interquartile range (IQR); N represents the number of patients in each group; PACU is the postoperative anesthesia care unit; POD is the postoperative day; *p*-value, significance level at 0.05. Summary statistics showing all IQRs of 0 due to discrete medication doses, low N comparisons, and low-frequency administration.

**Table 6 children-10-00193-t006:** Complications and medication side effects.

	Old Protocol/ Ropivacaine	New Protocol/ Ropivacaine Clonidine	*p*-Value
	N	N	
Nausea	26 (58)	8 (44)	0.3375
Pruritis	10 (22)	5 (28)	0.7457
Muscle spasm	16 (36)	11 (61)	0.0641
Urinary retention	4 (9)	1 (6)	1.0000
Respiratory depression	2 (4)	2 (11)	0.5712
Oversedation	0 (0)	1 (6)	0.2857
Constipation	6 (13)	2 (11)	1.0000
Other	2 (4)	3 (17)	0.1357

Legend: POD is the postoperative day; N represents the number of patients in each group; *p*-value, significance level at 0.05.

## Data Availability

UPMC Children’s Hospital, Department of Anesthesia archived dataset analyzed.

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
