# Peer review of "A Retrospective Study Comparing Outcomes of Paravertebral Clonidine Infusion for Pediatric Patients Undergoing Nuss Procedure"

_children, 2023, doi:10.3390/children10020193_

Round 1
Reviewer 1 Report
Pain study should demonstrate the pain score and pain medication consumptions. As the authors did indicate the pain score, however, the pain medication consumptions were shown in daily Total Morphine equivalents (mg/kg). In their protocol (old and new) PCA hydromorphone was given, then IV hydromorphone prn. Oral Oxycodone every 4 hours for pain. The ketamine infusion, acetaminophen, diazepam are included in the multimodal pain management protocol.
It will be more convincing to show each drug daily consumption between the two groups. As a retrospective study, more pain medication dose comparison data will strengthen the conclusion.
Since the surgery is done for pediatric population, did all patients use daily numerical rating scale (NRS) pain scores? Pediatric patients used different pain rating scale to report pain. Please clarify.
Author Response
Dear Reviewer #1
I appreciate your suggestions, responded to these concerns, and revised the manuscript accordingly. I believe these changes improved the quality of the manuscript. Unfortunately, I could not improve the research design. Thank you for considering my manuscript for publication in Children. My changes to the manuscript and answers to your questions are in red.
Your suggestion: It will be more convincing to show each drug's daily consumption between the two groups. As a retrospective study, more pain medication dose comparison data will strengthen the conclusion.
My answer: I added three more tables that described medication administration ( other than opioids) during surgery, in PACU, and over postoperative days 1- 6.
Your question: Since the surgery is done for the pediatric population, did all patients use daily numerical rating scale (NRS) pain scores? Pediatric patients used different pain rating scales to report pain. Please clarify.
My answer The median age for these patients was 15, and a numeric rating pain scale was used for all these patients.
Reviewer 2 Report
This paper is an interesting study about postoperative pain management with paravertebral clonidine infusion after Nuss procedure.
I would like to compliment the authors on their work.
Author Response
Dear Reviewer# 2
I appreciate your comments. Thank you!
Reviewer 3 Report
Nuss procedures have commonly been chosen as an area of study due to its painful course. As was mentioned by the authors, protocols are highly dependent on local practice so further study is warranted. I thank the authors for looking at their outcomes following Nuss procedures. I would implore the authors to continue to improve the patient experience.
Overall, the manuscript adds to the body of literature on postoperative pain control following these surgeries. However, so many changes were made between the two protocols that it is difficult to tease out the merits of clonidine as making the difference. Additionally, applicability of the knowledge gained from this manuscript may be limited. As was mentioned by the authors so few institutions are utilizing paravertebral blocks in their postoperative regimen.
Major points:
1. Authors add the point in line 246 that the outcome of reduced opioid use on POD 2 may be as a result of the entire protocol change which is significant. It would be challenging to attribute the reduced MME use to clonidine. The authors should consider revising the manuscript to show the change in protocol including clonidine may improve outcomes.
2. The data collected in this study is now well over 4 years old. Is there a reason why more recent patients were not included? Has the protocol changed since then?
Minor points:
1. Line 61- when did the protocol change? It would be helpful to see the timeline
2. Line 103- clarify oxycodone dosing
3. Line 113: clarify ibuprofen max dosing
4. Line 257: clarify clonidine dosing
Author Response
Dear Reviewer#3.
I appreciate your suggestions, responded to these concerns, and revised the manuscript accordingly. I believe these changes improved the quality of the manuscript, unfortunately. Thank you for considering my manuscript for publication in Children. My changes to the manuscript and answers to your questions are in red.
Your suggestion Authors add the point in line 246 that the outcome of reduced opioid use on POD 2 may be a result of the entire protocol change, which is significant. It would be challenging to attribute the reduced MME use to clonidine. The authors should consider revising the manuscript to show the change in protocol including clonidine may improve outcomes.
Thank you for this suggestion. I revised the discussion that says now: The decrease in opioid consumption on POD 2 may reflect overall changes in the new protocol, which included adding clonidine to the paravertebral infusion.
Your suggestion The data collected in this study is now well over 4 years old. Is there a reason why more recent patients were omitted? Has the protocol changed since then?
Thank you for this suggestion. I could include a few more patients in the clonidine group. However, I did not have good statistical support, and it took a while to get the data back. A few more things happened that delayed the submission for publication of this paper.
Our practice changed over the last three years. We are no longer using hydromorphone PCA for this population. We use almost the same protocol but offer ketamine PCA with continuous demand and Dilaudid as needed. We have surgeons that are using cryonyolysis. We still use paravertebral catheters with ropivacaine 0.2% and clonidine 1 mcg/ml for some of the Nuss procedures. For other surgeons, we do paravertebral blocks single injection, and for others, we do only medical management.
Line 61- when did the protocol change? It would be helpful to see the timeline.
We changed the protocol in 2016 and have been using clonidine in our paravertebral infusion since then. It is mentioned in the introduction. In 2016, we piloted a protocol that involved adding clonidine as an adjunct to ropivacaine infusion and removing continuous hydromorphone. These two pain management techniques are evolutions of each other.
Line 103- clarify oxycodone dosing.
I am sorry for the typo; I meant 0xycodone 0.1 mg/kg, maximum 5 mg. I corrected this. A maximum of 10 mg was added in the second paragraph.
Line 113: clarify ibuprofen max dosing
I am sorry for the deleted words after using the track changes option. Maximum of ibuprofen was 3200 mg/day..
Line 257: clarify clonidine dosing
I am sorry for the deleted number. It says now: paravertebral infusion of 0.125% bupivacaine at 0.5 mg/kg/h with clonidine at 2 mcg/kg/h.
Round 2
Reviewer 1 Report
the new table had missing data. for example Ketorolac (mg)
5 (11)
0.00 (0.00, 0.00)
6 (33)
0.00 (0.00, 15.00)
why the number is zero. the authors should explain to the readers
Author Response
Dear Reviewer#1,
Thank you for your suggestions. This happened because I reported the median and interquartile ranges. I checked with the statistician, and situations like this typically happen with discrete measures, like medication, that may not always be administered, especially in comparisons with low n (this occurs with n<10).
I did two things:
- A added a footnote under the tables that says: “Summary statistics showing all IQRs of 0, due to discrete medication doses, low n comparisons, and low-frequencyy administration..”
- In the methods section where the statistics, I added, “Due to discrete medication doses, low n comparisons and low-frequency administration of certain drugs, IQR ranges may show all zeros. This is not an error and shows that at least three-quarters of the comparable sample received no dose and suggests uncommon medication use in this particular study.My new changes are in green.
- Thank you!